# Genetic breakdown of a Tet-off conditional lethality system for insect population control

Yang Zhao[1,2], Marc F. Schetelig ● [3] & Alfred M. Handler ● [2✉]

Genetically modified conditional lethal strains have been created to improve the control of insect pest populations damaging to human health and agriculture. However, understanding the potential for the genetic breakdown of lethality systems by rare spontaneous mutations, or selection for inherent suppressors, is critical since field release studies are in progress. This knowledge gap was addressed in a *Drosophila* tetracycline-suppressible embryonic lethality system by analyzing the frequency and structure of primary-site spontaneous mutations and second-site suppressors resulting in heritable survivors from 1.2 million zygotes. Here we report that $F_1$ survivors due to primary-site deletions and indels occur at a $5.8 \times 10^{-6}$ frequency, while survival due to second-site maternal-effect suppressors occur at a ~$10^{-5}$ frequency. Survivors due to inherent lethal effector suppressors could result in a resistant field population, and we suggest that this risk may be mitigated by the use of dual redundant, albeit functionally unrelated, lethality systems.

[1] State Key Lab for Conservation and Utilization of Subtropical Agro-Biology Resources, Guangxi University, 100 Daxuedong Road, 530005 Nanning, Guangxi, China. [2] Center for Medical, Agricultural and Veterinary Entomology, USDA/ARS, 1700 SW 23rd Drive, Gainesville, FL 32608, USA. [3] Department of Insect Biotechnology in Plant Protection, Justus-Liebig University Gießen, Winchesterstr. 2, 35394 Gießen, Germany. ✉email: al.handler@usda.gov

Genetically modified (GM) conditional lethal insect strains have been developed to improve the sterile insect technique (SIT), with a focus on eliminating females during rearing for sexing, and genetic sterilization of males due to inviable progeny[1,2]. Two systems achieve non-sex-specific and female-specific conditional lethality by using the *Escherichia coli* tetracycline (Tet) resistance operon[3] to suppress expression (Tet-off)[4] of a lethal effector by feeding the antibiotic during rearing. One system, known as RIDL (release of insects carrying a dominant lethal), uses toxic overexpression of the tet-transactivator (*tTA*) in a self-promoting driver/effector cassette to effect lethality in the absence of Tet[5], while a Tet-off embryonic lethality system (TELS) uses a developmentally regulated *tTA* driver to promote the lethal effector expression of a pro-apoptotic cell death gene in embryos[6]. Both systems are highly effective in small-scale experimental studies in several insect species[7–10], and the RIDL system has advanced to field release studies for *Aedes aegypti*[11,12]. However, in laboratory control studies for RIDL, survival in the medfly, *Ceratitis capitata*, and the yellow fever mosquito, *A. aegypti*, were reported at frequencies of 0.5% and 3.5%, respectively[7,9], while survival for the *Drosophila melanogaster* TELS strain was approximately 0.01%[6]. These levels of initial $F_1$ survival are likely due, primarily, to inherent "leakiness" in the respective systems due to variable transgenic lethal effector expression or function, though heritable survival due to mutations in genetic components of the system have yet to be reported. This is due, in part, to laboratory screening of lethality lines being limited to thousands of adults while spontaneous mutations in *Drosophila*, resulting in loss-of-function alleles, have been reported in the range of $1–5 \times 10^{-6}$ per locus per generation[13]. Thus, if mass-reared GM conditional lethal strains are released in the field at current levels of $10^6–10^8$ insects/week for SIT[14,15], there is a high likelihood that rare primary- or second-site mutations will occur resulting in lethal revertant survivors. This would result in the persistence of GM insects in the field, and for some mutations (especially at second sites), revertants may be resistant to further control by the same or similar lethality system. Indeed, a recent report of *A. aegypti* populations in Brazil after the release of RIDL mosquitoes indicates introgression of RIDL strain genomic sequences in the surviving population[16], which could be associated with inheritance from non-mutant $F_1$ escapers or with inherent resistance to the tTA lethal effector allowing hybrid survival.

To experimentally assess the potential for breakdown of Tet-suppressible lethality, we have tested the frequency and genetic basis of heritable lethal revertant survivors for a *D. melanogaster* non-sex-specific TELS strain reared on Tet-free diet. Despite not being a significant economically important species, its use as a genetic model provides relevant knowledge relating to the frequency and structure of spontaneous mutations, and it is highly amenable to large-scale laboratory rearing. The binary driver/effector TELS system also provides the same primary *tTA* target site for mutation that is necessary for lethality in both the TELS and RIDL systems, thus allowing potential breakdown of both Tet-off conditional lethal systems to be evaluated within the same genomic context. In this respect, after screening approximately 1.2 million zygotes on Tet-free diet, 20 heritable lethal revertant survivors were discovered, 7 of which are due to primary-site spontaneous mutations in the driver/effector cassettes while the remaining 13 are due to second-site maternal effect suppressors. Primary-site survivors could still be controlled by the lethality system in the field, though second-site lethal effector suppressors are likely to be resistant. We suggest that the risk of insects resistant to the primary lethality system survive and expand in a field population may be mitigated by the concurrent use of a secondary redundant lethality system that is functionally independent.

## Results

**Evaluation of a Tet-off embryo-specific lethality line.** The TELS strain tested included the original *piggyBac* lethal effector vector, pB{3xP3-ECFP-5'HS4>5'HS4-TREp-hid^Ala5^-5'HS4>5'HS4} (line M5.II) as described by Horn and Wimmer[6] (Fig. 1a). This vector is comprised of a phospho-mutated *hid*^Ala5^ pro-apoptotic cell death lethal effector gene[17] linked to a tetracycline response element (TRE), surrounded by two duplicate pairs of 5' HS4 chicken β-globin insulator sequences[18], and the *3xP3-ECFP* marker gene (Fig. 1b). The *piggyBac* serendipity α-regulated *tTA* driver vector, pB{PUb-DsRed.T3, s1-tTA} (line M18A), was modified from the original embryonic *tTA* driver (*s2-tTA*)[6,19] by use of the *s1* promoter that has an additional 95 upstream promoter nucleotides (Fig. 1a) and includes the *polyubiquitin*-regulated *DsRed.T3* fluorescent marker (Fig. 1b). Both vectors were integrated into the second chromosome of independent *D. melanogaster white*⁻ mutant host strains, recombined for common linkage, and then inbred to create the double-homozygous driver/effector strain, DH-1.

The applied use of TELS in the field would require the mass release of double-homozygous $P_1$ males reared on Tet-diet, with the expectation that their double-heterozygous $F_1$ progeny would be inviable by the first larval instar in the absence of Tet. Thus $F_1$ survivors would be considered to be putative lethal revertants due to spontaneous primary-site point mutations, deletions or insertions in the driver or effector cassettes, or second-site modifiers or variations in the genome that would somehow suppress the otherwise functional TELS. However, fertile adult survivors expressing the DsRed driver (Red) and ECFP effector

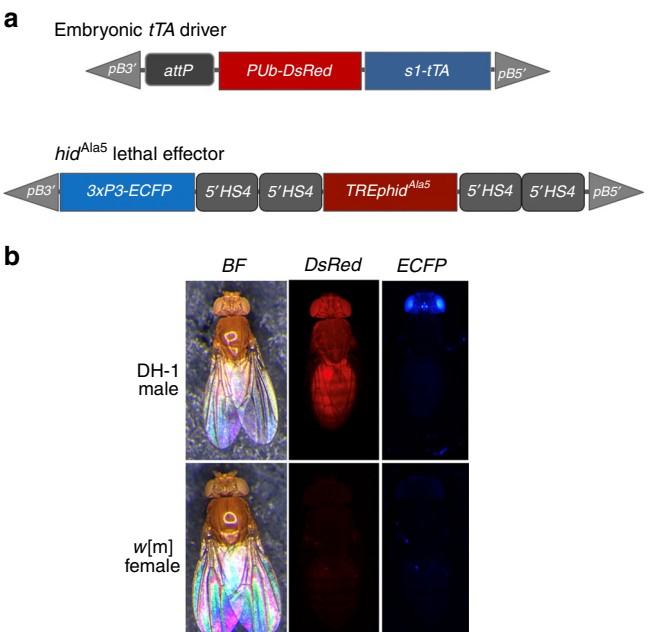

**Fig. 1 The *tTA* driver and lethal effector *piggyBac* vectors and phenotypes of a DH-1 male and *w*[m] female. a** Structure of the pBXL{attP_PUbDsRed.T3_fa_s1-tTA_a} *tTA* driver and pBac{3xP3-ECFPaf; 5'HS4>5'HS4>TREp-hid^Ala5^>5'HS4>5'HS4} lethal effector vectors in the double-homozygous line, DH-1. **b** Images of an adult DH-1 male and *w*[m] female under brightfield (BF) and Texas Red (DsRed) and CFP (ECFP) epifluorescence optics. DH-1 males, having a *w*[m] background, exhibit the *white*⁻ eye phenotype under brightfield and whole-body *PUbDsRed* and eye-specific *3xP3-ECFP* fluorescent markers for the driver and effector vectors, respectively. Non-transgenic *w*[m] females exhibit the *white*⁻ eye phenotype under brightfield and are non-fluorescent for both markers.

**Table 1 Tet-off lethality revertant adult survivors on Tet-free diet.**

| Matings | No. of $P_1$ matings[a] | Diet | $F_1$ R/C survivors | Fertile $F_1$ R/C survivors | $F_2$ R/C survivor lines[b] | Primary-site revertants | Second-site revertants |
|---|---|---|---|---|---|---|---|
| Control | 6 | Tet | 90,006[c] | — | — | — | — |
| Experimental | 80 | Tet-free | 109 | 73 | 20 | 7 | 13 |
| Survival frequency[d] | | | $9.1 \pm 0.45 \times 10^{-5}$ | $6.1 \pm 0.30 \times 10^{-5}$ | $1.7 \pm 0.083 \times 10^{-5}$ | $5.8 \pm 0.29 \times 10^{-6}$ | $1.1 \pm 0.054 \times 10^{-5}$ |
| Modified survival frequency[e] | | | $8.9 \pm 0.44 \times 10^{-5}$ | $5.9 \pm 0.29 \times 10^{-5}$ | $1.4 \pm 0.071 \times 10^{-5}$ | $3.3 \pm 0.17 \times 10^{-6}$ | $1.1 \pm 0.054 \times 10^{-5}$ |

[a]400 w[m] females mated to 100 DH-1 double homozygous driver/effector (PUb-DsRed/3xP3-Cyan; R/C) males.
[b]$F_2$ R/C progeny from individual fertile $F_1$ R/C backcross matings to w[m].
[c]Calculated from 15,001 ± 389 SEM surviving $F_1$ adults per control mating.
[d]No. of R/C marked $F_1$ survivors/~1,200,000 ±SEM adults screened (estimation based on surviving $F_1$ adults from each control mating on Tet-diet).
[e]Modified survival frequency data based on the deletion of the primary-site revertant lines 9-f1, 62-f1, and 65-m2 whose survival may have resulted from cis-recombination between the 5' HS4 repetitive insulator sequences that are not required for lethality system function, which does not affect the second-site revertant frequency; frequencies presented as confidence intervals at the 95% confidence level.

(Cyan) cassette markers would require verification as true heritable lethal revertants from a backcross to w[m] that yields double-marked Red/Cyan $F_2$ surviving progeny on Tet-free diet (Fig. 1b).

**$F_1$ survivors from large-scale rearing.** Based on typical spontaneous mutation rates of $10^{-6}$ to $10^{-5}$ per locus in Drosophila[13], we presumed that the recovery of a significant number of heritable lethal revertants from the two loci TELS strain would require a minimal screen of ~$10^6$ $F_1$ adults from large-scale rearing. This was achieved by establishing 6 control mating groups, each having 100 homozygous DH-1 males crossed to 400 w[m] virgin females reared on permissive Tet-diet, with two successive transfers to fresh media, which yielded a total of ~15,000 $F_1$ adult progeny from the three broods for each mating group (Table 1). The same crosses were then established for 80 experimental mating groups reared on non-permissive Tet-free diet that would account, based on control matings, for the screening of approximately 1,200,000 double-heterozygous $F_1$ progeny for adult survival. From the experimental matings, a total of 109 Red/Cyan (R/C) double-marked adults, for the driver and lethal effector vector cassettes (Fig. 1b), eclosed on Tet-free diet yielding an initial survival rate of 0.0091% (Table 1), consistent with preliminary small-scale population survival rates.

**Heritable $F_2$ lethal revertants identified from $F_1$ survivors.** From the 109 $F_1$ survivors, 88 remained viable allowing backcross matings to w[m] to test for heritability of disruption or suppression of the lethality system (Table 1). Seventy-three $F_1$ survivors were fertile, of which 20 resulted in R/C $F_2$ progeny on Tet-free diet, indicating a heritable survival frequency of 18.3% relative to the initial number of survivors, and a survival frequency of $1.67 \times 10^{-5}$ relative to the total number of adults screened (Table 1). Seven of the 20 survivor lines exhibited an approximate 1:1 ratio of unmarked to double-heterozygous R/C marked $F_2$ descendants on Tet-diet, with approximately 11% of the progeny expressing either the R or C marker (Table 2). Independent marker segregation was presumably due to recombination between the driver and lethal effector cassettes at their respective chromosomal 2R loci of 56C8/9 and 53C4 (Supplementary Fig. 1), limited to the five $F_1$ female lines due to the absence of male recombination in D. melanogaster[20]. These results were consistent with complete disruption of the lethality system due to spontaneous primary-site genetic alterations in either, or both, the tTA driver or lethal effector cassettes at a frequency, relative to the population screened, of $5.8 \times 10^{-6}$ events per generation. The significance of this frequency is supported by a power calculation[21] to test for the optimal sample size required for detecting primary-site genetic alterations based on the typical null mutation rate of $10^{-6}$. This resulted in a power of 0.93 at the

0.05 significance level for the screened sample size of ~1,200,000 double-heterozygous $F_1$ progeny (Supplementary Fig. 2).

**Second-site maternal effect lethality suppression lines.** The 13 additional $F_2$ R/C survivor lines were heritable but exhibited significantly lower survival frequencies for R/C marked progeny relative to their unmarked siblings (Table 2), ranging from 1.4% to 32.4% survival, with 11 lines being <11%. These variable R/C frequencies are more consistent with lethal suppression by second-site modifiers, either distantly or un-linked to the driver/effector cassettes, and possibly the result of different suppressors.

Notably, all of these lines originated from $F_1$ females, although $F_2$ survivors included both sexes, suggesting the possibility of maternal effect heritability of lethal suppression. This was tested by backcrosses of R/C revertant males and females to w[m] flies in successive generations. For all 13 lines, $F_3$ R/C survivors only arose on Tet-free diet from maternal R/C $F_2$ females, and not from R/C $F_2$ males, while both $F_2$ males and females gave rise to $F_3$ R/C progeny on Tet-diet. For two lines (5-f1 and 14-f2) having the highest levels of survival, the maternal effect was observed for 11 generations and confirmed by 10 to 12 individual backcross matings of $F_{10}$ males and females to w[m] (Table 3).

**Molecular analysis of lethal revertant survivor lines.** To determine the molecular integrity of the tTA driver and lethal effector cassettes in the lethal revertant survivor lines, $F_2$ R/C adults for each line were inbred to homozygosity on Tet-free diet. PCR products from the s1-tTA driver and TREp-hid[Ala5] lethal effector cassettes from the revertant lines were sequenced (Supplementary Table 2 and Supplementary Fig. 3) using adjacent and internal primers, though reliable PCR amplification of the highly repetitive 5' HS4 insulator sequences was not possible. For some difficult sequences, more distant primers or TAIL-PCR were utilized. For the seven putative primary-site revertant lines, relatively short deletions were identified within the tTA and hid[Ala5] genes, in addition to larger deletions and indels that included TREp-hid[Ala5] and adjacent insulators (Fig. 2). Notably, mutations or alterations were not evident in the driver or effector cassettes for any of the 13 putative modifier lines, consistent with second-site mutations or variations.

For the primary-site mutations, two relatively short spontaneous deletions were detected in lines 18-f1 and 45-f1. In line 18-f1, the TREp-hid[Ala5] lethal effector sequence was unaltered, but a 27-bp deletion (nts 221–247 from the ATG) and three adjacent nucleotide substitutions were identified in the tTA driver coding sequence (Fig. 2 and Supplementary Fig. 4a). While the tTA reading frame was maintained, we presume the deleted and substituted peptides resulted in loss of tTA function. In line 45-f1, the tTA driver was unaltered, but a 26-bp deletion (nts 27–52 from the ATG) occurred in the 1.2-kb hid[Ala5] lethal effector gene,

**Table 2 Tet-off revertant lethal F$_2$ and F$_3$ survivors reared on Tet-free diet.**

| Survivor lines | w[m] backcross F$_2$ survivors | % unmarked | % Red and Cyan (revertant) | % Red or Cyan (recombinant) | F$_3$ maternal effect survivors[a] |
|---|---|---|---|---|---|
| Primary site | | | | | |
| 9-f1 | 209 | 46.4 | 43.1 | 10.5 | − |
| 18-f1 | 212 | 45.8 | 42.0 | 12.2 | − |
| 21-f1 | 251 | 52.2 | 39.4 | 8.4 | − |
| 44-m1 | 91 | 46.2 | 53.8 | 0 | − |
| 45-f1 | 244 | 50.0 | 39.8 | 10.2 | − |
| 62-f1 | 197 | 47.7 | 40.1 | 12.2 | − |
| 65-m2 | 117 | 54.7 | 45.3 | 0 | − |
| Second site | | | | | |
| 2-f1 | 107 | 78.5 | 1.9 | 19.6 | + |
| 5-f1 | 105 | 54.3 | 32.4 | 13.3 | + |
| 14-f1 | 86 | 75.6 | 3.5 | 20.9 | + |
| 14-f2 | 72 | 70.8 | 16.7 | 12.5 | + |
| 16-f1 | 55 | 72.7 | 10.9 | 16.4 | + |
| 26-f1 | 12 | 75.0 | 8.3 | 16.7 | + |
| 27-f1 | 151 | 77.5 | 4.6 | 17.9 | + |
| 49-f1 | 69 | 81.2 | 1.4 | 17.4 | + |
| 55-f1 | 58 | 74.1 | 8.6 | 17.2 | + |
| 69-f1 | 67 | 76.1 | 3.0 | 20.9 | + |
| 70-f1 | 117 | 80.3 | 5.1 | 14.5 | + |
| 73-f1 | 128 | 79.7 | 2.3 | 18.0 | + |
| 79-f1 | 71 | 77.5 | 2.8 | 19.7 | + |

[a](−) absence of maternal effect heritability based on F$_3$ R/C surviving progeny from both R/C F$_2$ females and R/C F$_2$ males backcrossed to w[m]; (+) maternal effect heritability based on F$_3$ R/C surviving progeny only from backcrossed R/C F$_2$ females backcrossed to w[m] males and not from backcrossed R/C F$_2$ males.

**Table 3 Revertant lethal line F$_{10}$ females or males backcrossed to w[m] exhibiting maternal effect F$_{11}$ survival on Tet-free diet.**

| Revertant line | F$_{10}$ sex | No. of matings[a] | Tet-diet | Unmarked F$_{11}$ survivors | Red/Cyan F$_{11}$ revertant survivors | Red or Cyan F$_{11}$ recombinants[b] |
|---|---|---|---|---|---|---|
| 5-f1 | F | 10 | + | 539 | 461 | 175 |
| | F | 12 | − | 584 | 243 | 143 |
| | M | 11 | + | 641 | 644 | 0 |
| | M | 12 | − | 928 | 0 | 0 |
| 14-f2 | F | 11 | + | 505 | 505 | 177 |
| | F | 12 | − | 556 | 337 | 150 |
| | M | 11 | + | 679 | 683 | 0 |
| | M | 12 | − | 860 | 0 | 0 |

F virgin female, M male.
[a]One revertant line parental mated to three w[m] adults of the opposite sex per mating.
[b]Note that recombination does not occur in D. melanogaster males.

resulting in an early frameshift that likely eliminated gene product function (Fig. 2 and Supplementary Fig. 4b).

For the primary-site lines 21-f1 and 44-m1, the *tTA* driver sequence was unaltered, but a series of PCR primer pairs flanking or internal to the *5'HS4_5'HS4-TREp-hid*$^{Ala5}$*-5'HS4_5'HS4* lethal effector cassette failed to yield amplicons, consistent with a large deletion or indel (see "Methods" and Supplementary Fig. 3b, c2–7). Control PCRs were positive for the unaltered presence of the *pB3'* to *3xP3-ECFP* sequence (AH1577–AH1636) in both lines (Supplementary Fig. 3b, c1) consistent with the expression of the marker gene. Internal PCR (AH1540–AH1652) indicated the presence of the effector *pB5'* sequence in both lines (Supplementary Fig. 3b, c8); however, PCR from *pB5'* to adjacent 3' genomic sequences were not positive (Supplementary Fig. 3b, c6) suggesting a transposition or rearrangement of sequences. Partial insert sequences downstream from *ECFP-SV40* were identified by TAIL-PCR, which included a 180-bp incomplete insert of unknown origin in line 21-f1 (Supplementary Fig. 5a), except

for several short sequences identical to those found on the X and 2R chromosomes, the *3xP3* promoter, and a *5' HS4* insulator. Line 44-m1 revealed a 540-bp incomplete insert (Supplementary Fig. 5b) having high identity to telomeric HeT-A non-LTR retrotransposon sequences, with 87% identity to the X chromosome HeT element (ID: M81595.1)[22] (Supplementary Fig. 6). Notably, retrotransposon insertions, among other mobile elements, have been associated with deletions that may provide the breakpoints for their insertion resulting in spontaneous mutations in *Drosophila* and other eukaryotes[23,24].

For the three remaining heritable survivor lines (9-f1, 62-f1, and 65-m2), PCR with primers in *ECFP* and *pB5'* and *ECFP*-linked SV40 and *pB5'* yielded 2.9 kb (Supplementary Fig. 3c7) and 2.6 kb (Supplementary Fig. 7a) products, respectively, that included partial *5' HS4* insulator sequences and the absence of the internal *TREp-hid*$^{Ala5}$ sequence. The actual sequence in this region was then inferred by *Afl*II and *Bgl*II digests of the 2.6-kb product that resulted in fragment sizes consistent with a deletion

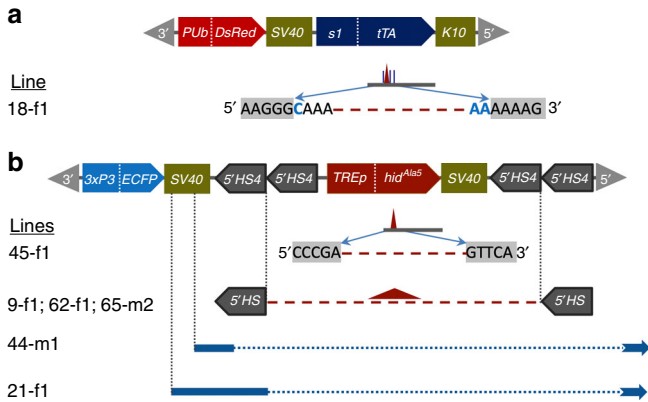

**Fig. 2 Genetic alterations in the *tTA* driver and lethal effector vectors in primary-site lethal revertant survivor lines. a** Line 18-f1 has a 27-bp deletion and 3 nucleotide substitutions (blue vertical lines and nucleotides) in the *tTA* gene and **b** line 45-f1 has a 26-bp deletion in the *hid*$^{Ala5}$ cell death gene (maroon dashes and triangles indicate deleted sequences). Lines 9-f1, 62-f1, and 65-m2 have identical deletions of the *TREp-hid*$^{Ala5}$ lethal effector and two 5′ *HS4* insulator sequences, lines 44-m1 and 21-f1 exhibit deletions (blue dotted lines), and insert sequences (blue solid lines) from within the *ECFP*-linked *SV40* extending downstream through the 3′ genomic insertion site sequence (blue arrows indicate unknown indel termination sites).

of the lethal effector and two insulators (Supplementary Fig. 7a). If the two remaining insulator sequences were originally upstream and downstream of *TREp-hid*$^{Ala5}$, a parsimonious explanation for the deletion would be a single-strand inverted-loop meiotic pairing of the duplicate insulator tandem pairs having a single internal recombination event. The resulting segregation products would thus consist of the two remaining insulators due to a circular DNA deletion of 5′HS4_TREphid$^{Ala5}$_5′HS4 (Supplementary Fig. 7b).

## Discussion

After nearly 20 years since Tet-off conditional lethality systems for pest insect population control were first reported[5,6,25], a quantitative and structural assessment for the spontaneous breakdown of the genetic components that comprise the *D. melanogaster* Tet-off embryonic lethality system has now been described. After screening approximately 1.2 million heterozygous TELS F$_1$ adults on a restrictive Tet-free diet, 20 heritable revertant lethal survivors were recovered, yielding a frequency of $1.7 \times 10^{-5}$ events per generation resulting in genetic breakdown. However, primary-site mutations disrupting the *tTA* driver or *hid*$^{Ala5}$ lethal effector sequences were verified for seven of the survivor lines that included deletions and indels of various length, including a retrotransposon-associated indel, resulting in a spontaneous primary-site mutation rate of approximately $5.8 \times 10^{-6}$ events per generation.

Given that three of the lethal revertant lines were likely due to deletions resulting from *cis*-recombinations between the highly repetitive 5′ *HS4* chicken globin insulators, a more relevant revertant frequency that includes only the remaining four mutant lines would be ~$3.3 \times 10^{-6}$ events per generation (Table 1). Since this mutation rate includes genetic alterations from both primary-site targets, it more accurately reflects the overall rate of genetic breakdown for this specific lethality system, whereas the approximate spontaneous mutation rate per locus for the 1.0-kb *tTA* driver and 1.2-kb *hid*$^{Ala5}$ lethal effector coding regions would be $0.83 \times 10^{-6}$ and $2.5 \times 10^{-6}$ events per locus, respectively. Various factors can affect the mutation rate of specific loci, including the method of phenotypic screening and, indeed, our

revertant lethal screens depended on detecting dual DsRed/ECFP marked survivors that would have missed revertants due to deletions including critical sequences for either marker gene. With these caveats in mind, the observed spontaneous mutation rates for the driver and effector alleles remain generally consistent with previous spontaneous mutation studies for a variety of alleles in *Drosophila*, having a mean spontaneous specific locus rate of $1–5 \times 10^{-6}$ as estimated by Ashburner[13] and additional analyses[24,26–29].

The recovery of putative insulator recombinations also highlights an important consideration for the design of lethality systems since the insulator pairs surrounding both driver and effector cassettes in the original TELS strain did improve lethal gene expressivity[6]. However, this was achieved at the cost of increasing the primary-site mutation frequency by >40% (Table 1). Thus an important caveat for the use of the 5′ *HS4* insulator, and possibly other repetitive sequences, is the possibility for internal *cis*-recombinations resulting in increased lethality strain breakdown due to deletions in a significant percentage of F$_1$ progeny.

The mutation recovered in this study most relevant to the RIDL system, for which *A. aegypti* RIDL lines have already been tested in open field release studies[9,30], was the short internal deletion within the *tTA* driver that acts as both driver and lethal effector in the self-promoting RIDL system. Given the $0.83 \times 10^{-6}$ mutation frequency of this locus from a single F$_1$ TELS revertant, it is possible to expect a RIDL insect to survive from at least every 1.2 million F$_1$ embryos oviposited in the field. Not unexpectedly, the additional *hid*$^{Ala5}$ effector cassette critical for TELS function, compared to RIDL, provides an additional target for mutation and thus a higher frequency for disruption of lethality based on this study.

In addition to primary-site mutations, this study revealed a class of second-site maternal effect variations, found in 13 of the revertant lethality lines, that effectively suppress expression or function of the *hid*$^{Ala5}$ pro-apoptotic lethal effector gene. While it remains to be determined how ectopic *hid*$^{Ala5}$ could be negatively influenced by a maternally derived factor, there is precedence for a maternally inherited mitochondrial role in pro-apoptotic gene product function. In mammalian systems, cytochrome c is converted into holocytochrome c by a heme attachment within the mitochondrial intermembrane space that facilitates pro-apoptotic caspase activation resulting in cell death[31]. While this specific function has not been observed in invertebrates, a mitochondrial role in *Drosophila* cell death has been inferred by the inhibition of apoptosis when pro-apoptotic protein localization within the mitochondria is blocked[32]. Although the observed maternal effect may be limited to the particular *D. melanogaster* w[m] host strain tested, if ectopic pro-apoptosis is similarly subject to suppressor gene function due to variation in a small subset of an insect population, then the potential for selection of these suppressors should be a concern for the use of pro-apoptotic lethal effectors. Indeed, the *hid* gene has already been tested as a TELS conditional lethal effector in four insect pest species[8,10,33–35], some of which are in consideration for release.

Another mechanism for maternal effect suppression, since the *sry-alpha* promoter functions transiently in early embryogenesis[19], is a pre-zygotic maternal contribution of molecules that suppress ectopic *tTA* or *hid*$^{Ala5}$ expression or function at that developmental period. Since the RIDL system relies on the toxic accumulation of tTA protein throughout development for lethality, neither of these types of maternal effect suppression may be relevant to RIDL, though it is possible that tTA toxicity ultimately results in induced apoptosis.

More broadly, however, is the realization that regardless of the lethal effector employed for population control, or its mode of

action, it may be subject to suppression by a pre-existing inherent variation in the targeted field population[36] or a newly acquired modifier mutation in the mass-reared lethality strain. In either case, such effects could have a significant impact on lethality system effectiveness and implications for its use in population control programs. Importantly, this would require evaluation by contained large-scale preliminary testing as performed in this study but, ideally, using field population adults for $P_1$ matings. Another important consideration is that primary-site revertant survivors, due to *cis*-acting mutations, of both TELS and RIDL systems should still be susceptible to control by the same lethality system. In contrast, a dominant-acting second-site suppressor of lethal effector expression or function, due to *trans*-acting modifiers, would more likely result in GM insects resistant to the lethality system, and relatively small numbers of GM resistant survivors could be expected to rapidly re-populate an ecosystem where the susceptible population has been suppressed. Certain factors will affect the rate of introgression of the resistant population and its ability to reach fixation. These include the number of suppressor alleles selected for and whether they are recessive lethal, recessive or dominant fitness costs, maternal effect heritability, and the proportional population density of successive releases among other variables having positive or negative effects[37]. All of the TELS suppressor lines were inbred to homozygosity (and share the same mitochondrial genome), and none exhibited recessive lethality or apparent semi-lethality due to compromised fitness. Nevertheless, unless the negative costs to resistance are considerable, the eventual establishment of a lethality-resistant population may be expected from programs that rely on continuous mass releases (especially preventative release programs[38]) in the absence of an efficient secondary control program. The potential for resistance to bi-sexual and female-specific RIDL release programs, and its spread within a population, has been modeled taking into account many of the variables affecting resistance introgression for this system[37], but this remains theoretical in the absence of large-scale population tests and knowledge of the actual basis for tTA lethality. The importance of this knowledge for the RIDL system, in particular, is heightened by recent studies showing significant introgression of genomic sequences from the *A. aegypti* OX513A RIDL strain into the native Brazilian (Jacobina) mosquito population 6 months to 2.5 years after the initiation of lethal strain releases[16]. Notably, sequences related to the lethality system were not reported, nor could they necessarily be identified by the single-nucleotide polymorphism analysis performed, and the sequences identified could have simply been inherited from non-mutant escapers that comprised >70% of the $F_1$ TELS survivors in our tests. But until this is resolved, one explanation for at least some of the observed introgression is survival due to inherent resistance to the tTA lethal effector. Alphey et al. suggest early and effective monitoring of release programs to detect resistance in time for remediation[37], which is certainly important on a continuing basis, as it is for most current SIT programs[38]. However, the additional use of preliminary large-scale contained population tests previous to a new release program would provide the advantage of assessing the potential for resistance and a determination as to whether concurrent or sequential releases of a secondary control system, or other modifications, would be necessary.

Considering the possibility that resistance to current insect conditional lethality systems might occur, strategies to prevent or mitigate the survival of lethality-resistant individuals have been considered. Two strategies are based on the potential for redundant, yet independent, dual lethality (or reproductive sterility) systems mitigating the survival of revertant individuals in a mass release program[39,40]. Thus, based on the multiplicative law of probability $[P(AB) = P(A)P(B)]$, an approximate $10^{-6}$ lethal revertant frequency estimate for each system ($A$ and $B$), as demonstrated here for a *Drosophila* TELS strain, could result in a frequency as low as $\sim 10^{-12}$ for both to fail concurrently in the same genome, presuming that the lethality systems are functionally independent and do not differentially affect host physiology and fitness. In application, at a modest rate of release for current SIT programs ($\sim 10^7$ males per week), use of a single lethality system might result in ≥100 heritable revertant survivors per week assuming each male yields 10 $F_1$ progeny, whereas the coincident breakdown of a dual lethality strain might yield a single revertant survivor only after several decades.

From these considerations, we would suggest that large-scale population screens for spontaneous mutations or pre-existing genetic variations, resulting in the genetic breakdown of conditional lethal systems developed for mass field release, be a critical component in the risk assessment evaluation of these programs. This could be achieved most efficiently in rearing facilities where mass-reared $P_1$ males and females are mated under non-permissive conditions for survival, whereupon the frequency and structure of mutations and suppressor/modifiers could be evaluated in surviving $F_1$ progeny. Given that most current facilities for SIT mass-rearing generate a minimum of $10^7$–$10^8$ insects per week, a more significant and informative analysis could be achieved in insects of interest relative to large-scale laboratory studies with *Drosophila*. Importantly, host females for $P_1$ matings should come from the targeted field populations to test for inherent suppressors or modifiers of the lethality system as a prelude to actual releases, though such evaluations may prove to be less critical with the use of redundant lethality systems.

This study has focused on a conditional lethal strain created by transposon-mediated transformation, yet most insect genetic modifications should be subject to the same types of alteration, disruption, or suppression regardless of the methodology used to create them. However, the frequency of specific types of mutations and suppressor/modifiers may vary with the genomic structure of the host species (e.g., genome size, chromatin structure, repetitive DNA content, etc.) and the specific modification. This is especially relevant to the variety of manipulations envisioned by gene editing, and the development of gene drive (GD) systems in particular. Indeed, despite the great potential for GD systems to modify or replace highly pestiferous insect strains in the field, it is well recognized that control or containment systems must be integrated into GD systems before the field release of these self-propagating strains can be considered[41,42]. A variety of containment strategies proposed, and in some cases tested, include drives that are self-limiting[43] such as multi-component daisy-chained drives[44], split drives[45,46] and spatially restricted drives[45] and small molecule-dependent drives[47], among others. However, similar to the lethality strain breakdown demonstrated here all of these strategies should be subject to mutation and modification and, potentially, at higher frequencies due to multi-generational GD strategies and larger target sites for some. Thus it would seem imperative that similar large-scale studies for GD integrity be initiated for systems already created in *Drosophila*, and if warranted, redundancy for GD containment be considered and tested for mitigation. Similarly, the desired systems for introgression should be equally subject to genetic breakdown for which redundancy may also be required, at the risk of a GD population regaining its pestiferous function, or losing its beneficial function, and expanding in nature. In this respect, and in general, care should be taken in the design of all genetic modifications for specific applications, in terms of understanding and testing the potential for mutation, suppression, and the possible need for redundancy to maintain strain stability.

## Methods

**Insect strains, rearing, and screening**. All *D. melanogaster* transgenic lines, the white-eyed *w*[m] mutant line, and large-scale matings were reared under standard laboratory conditions at 25 °C and 60% humidity on a 12-h light:12-h dark cycle[48]. Flies were anesthetized with $CO_2$ for screening, fly collections, and matings. Transgenic flies were screened by epifluorescence microscopy using a Leica MZFLIII fluorescence microscope and filter sets, TxRed for DsRed (ex: 560/40; em: 610 LP) and CFP for ECFP (ex: 436/20; em: 480/40). The University of Florida Institutional Biosafety Committee provided oversight and regulatory approval for the creation and rearing of the GM *D. melanogaster* strains used in this study.

**Tet-off *tTA* driver and *hid*[Ala5] lethal effector lines**. The *D. melanogaster* transgenic embryonic *tTA* driver M18A strain was transformed with the *piggyBac* transgene vector, pBXL{attP_PUbDsRed.T3_fa_s1-tTA_a} (GenBank accession no.: MT453110). The vector was created by isolating the *s1-tTA* driver sequence, as an *Asc*I fragment, from pBac{3xP3-EYFPafm; s1-tTA}[6] and inserting it into the *Asc*I site in pBXL{attP_PUbDsRed_fa}[10]. Unlike the *tTA* driver vector used in the original embryonic lethality strain[6], the *serendipity α s1* promoter has an additional 95 upstream nucleotides compared to the s2 promoter, the 5' *HS4* insulator sequences that surround the *s2-tTA* driver construct are eliminated, an *attP* recombination site is added for subsequent phiC31-mediated integrations, and a *polyubiquitin*-regulated *DsRed* (PUbDsRed) whole-body marker replaces the eye-specific *3xP3-EYFP* marker. The vector was transformed into the *w*[m] strain by *piggyBac*-mediated germline transformation[49] using a 500 ng/µl transgene vector plasmid and 200 ng/µl *phspBac* transposase helper plasmid mixture for preblastoderm injections.

The transgenic M5.II lethal effector strain, pBac{3xP3-ECFPaf_5'HS>5'HS>TREp-hid[Ala5]>5'HS>5'HS4} was created and kindly provided by Horn and Wimmer[6]. This effector vector is marked with the eye-specific marker, *3xP3-ECFP*, and carries a Tet-suppressible lethal effector construct having the phospho-mutated *hid*[Ala5] pro-apoptotic cell death gene linked to a TRE. In the absence of tetracycline, the tet-transactivator (*tTA-VP16* originally isolated from the pTet-Off Vector plasmid; Takara BIO USA, Inc.) from the driver construct binds to the *tetO* operator sequences within the TRE to promote *hid*[Ala5] transcription. Included within the vector construct is an *attP* recombination site for subsequent phiC31-mediated integrations, and two tandem repeat 2.4 kb 5' *HS4* chicken β-globin insulator repeat sequences flank both sides of the effector construct.

**Vector insertion site mapping**. The effector cassette in strain M5.II was previously mapped to chromosome 2 but not to a specific locus[6]. Segregation patterns from the mating of the M18A and M5.II homozygous strains and subsequent inbreeding suggested that the effector cassette in M18A also maps to chromosome 2. Therefore, the M18A and M5.II homozygous strains were inter-mated and double-heterozygous progeny were outcrossed to *w*[m] to select recombinant progeny having the driver and effector cassettes linked on the same chromosome. Recombinants were inbred to create the double-homozygous driver/effector line, DH-1, that was mated to the *D. melanogaster white* mutant strain *w*[m] in large-scale rearing (Fig. 1).

Both the effector and driver cassettes in DH-1 were mapped to specific loci by inverse PCR insertion-site sequencing to one arm of each vector. Inverse PCR[10] was performed by digestion of the DH-1 strain with *Ava*II or *Rsa*I and amplifying circularized fragment sequences with primer sets AH1537–AH1579 for the driver cassette 3' flanking sequence and AH1566–AH1542 for the effector cassette 5' flanking sequence (Supplementary Table 2). The flanking genomic sequences were subjected to a BLASTn search of the *D. melanogaster* genome assembly indicating an insertion of the driver cassette at the *Tab2* gene (CG7417) and the effector cassette at the *Sema2b* gene (CG33960), both on chromosome 2R (Supplementary Fig. 1). However, while both transgenes inserted into a TTAA site (duplicated upon insertion) consistent with all *piggyBac* insertions, the genomic sequences immediately flanking both the 5' and 3' *piggyBac* termini were not present in the *D. melanogaster* NCBI database. The driver cassette 5' flanking sequence and the effector cassette 3' flanking sequence were then re-confirmed by direct PCR using primer sets, AH1536–AH1585 and AH1543–AH1570 (Supplementary Table 2), to internal vector and genomic sequences and were also confirmed by direct internal sequencing in the genomes of *w*[m] and another *white* strain, *w*[1118]. A straightforward explanation for the anomalous sequences is that many *white* strains have been maintained as inbred laboratory colonies for many years (at least 60 years for *w*[1118]) and may have been subjected to genetic drift from the wild-type strains used for recent whole-genomic sequencing.

**Tetracycline tests**. To generate an embryonic lethality strain exhibiting the lowest survival frequency on Tet-free diet and the highest level of survival on the lowest Tet-diet concentration (considering potential fitness costs at high concentrations), $F_1$ survival of progeny from small-scale matings of DH-1 males and *w*[m] females was first tested on Tet-free diet and diet supplemented with tetracycline at concentrations of 10, 50, and 100 µg/ml (Supplementary Table 1). Eggs were collected on medium containing the corresponding Tet concentration, with pupae and adult eclosion rates recorded. Since no significant difference was observed in eclosion

rates between 10 and 50 µg/ml, a concentration of 20 µg/ml tetracycline was used in media for large-scale rearing tests.

**Large-scale rearing tests**. For large-scale rearing tests, *w*[m] virgin females and double-homozygous DH-1 males were collected <8 h after eclosion and reared separately for 2 days before $P_1$ mating of 400 females and 100 males for brood I in 1-liter jars on 20 µg/ml Tet-diet for control tests and Tet-free media for experimental tests. Brood I adults were transferred to fresh media after 4 days for brood II, with another transfer after 4 days for brood III, with brood III adults discarded after 6 days. Six control matings were performed to determine the mean total number of surviving adults from the three broods for each mating, which were counted individually. This resulted in a mean number of 15,001 ± 389 standard error of the mean $F_1$ adults per mating (Table 1). A total of 80 experimental matings (240 jars) were screened for pupal survivors that were collected and placed in individual vials with Tet-free media and monitored for adult eclosion. $F_1$ adults were counted and screened for the fluorescent markers, and survivors double-marked for the driver (DsRed) and effector (ECFP) cassettes were backcrossed individually to *w*[m] adults on Tet-free diet for 3–5 days. $F_1$ female survivor matings were then transferred to Tet-diet for 4–5 days, while three *w*[m] females mated to $F_1$ male survivors were left on Tet-free diet and three females were transferred to Tet-diet. Rearing on Tet-diet ensured that a line from fertile $F_1$ adults was maintained and provided a control for the expected percentage of Red/Cyan $F_2$ progeny. If $F_2$ R/C progeny survived on Tet-diet, but not on Tet-free diet, then the $F_1$ individual was considered to be a non-heritable survivor and the line was discarded, but if the $F_2$ R/C progeny survived on Tet-free diet, it was considered to be a putative heritable lethal revertant survivor. $F_2$ R/C males and females were inbred to create double-homozygous lines by single sib-pair matings with selection of the strongest fluorescing progeny (presumed to be homozygous) for matings in successive generations until all progeny were strongly fluorescent, typically by the third generation. Homozygosity was then verified by backcrosses to *w*[m] that resulted in 100% heterozygous progeny.

**Heritable lethal revertant survivor lines**. Double-heterozygous $F_2$ progeny, expressing both DsRed and ECFP (R/C) fluorescent markers, from backcrossed $F_1$ survivors reared on Tet-free diet, were considered to be lethal revertants due to a heritable mutation or variant modifier allele. Primary-site mutations in either the driver or effector sequences, causing breakdown of the lethality system, were expected in approximately one-half of the $F_2$ progeny with the remaining progeny consisting of unmarked *w*[m] flies and single-marked progeny resulting from recombination between the markers from $F_1$ female survivors (since recombination does not occur in *D. melanogaster* males). Second-site modifiers were expected to have a dominant-acting function and would also be expected to suppress lethality in one-half of the double-heterozygous $F_2$ progeny only if the alleles had relatively close second chromosome linkage to the driver/effector cassettes and were fully expressed. Lower frequencies of $F_2$ survival would be expected if second-site modifiers were unlinked or distantly linked to the D/E cassettes or were variably expressed possibly due to independent suppressor alleles. $F_1$ and successive generation backcrosses on Tet-diet provided a control for the expected frequencies of marked and unmarked $F_2$ progeny. Heterozygous lethal revertant survivor lines were maintained by continuing backcrosses to *w*[m] on Tet-diet, in addition to inbred homozygous lines maintained on Tet-free diet for molecular analysis.

Putative second-site survivor lines, based on low $F_2$ survival frequencies, were tested for putative maternal effect modifiers by three replicate single pair backcrosses of $F_2$ double-heterozygous males and females to *w*[m] reared on Tet-free diet for 3 days and then Tet-diet for 3 days. Adult progeny on each diet were screened for markers, with the ratio of double-marked to unmarked $F_3$ descendants compared between the two diets. $F_3$ double-heterozygous survivors arising only from $F_2$ females, and not from $F_2$ males, was considered to be consistent with a second-site maternal effect modifier that was verified by continued backcrosses in successive generations and verifying that the sequence integrity of the driver and effector cassettes was unaltered (Table 2 and Supplementary Fig. 3a, b). For two putative maternal effect lines, 5f1 and 14f2, exhibiting relatively higher frequencies of survival, backcrosses were continued for 10 generations at which time 10–12 replicate single pair backcrosses of both $F_{10}$ double-heterozygous males and females was performed (Table 3).

**Molecular analysis of lethal revertant survivor lines**. The *s1-tTA* driver and *TREp-hid*[Ala5] lethal effector cassette sequences were isolated by PCR for all heritable homozygous survivor lines and sequenced to identify genetic alterations that would eliminate or diminish lethal effector expression. Primer pairs AH1601–AH1631 and AH1554–AH1582 were used to amplify the driver cassette, while the primer pairs AH1616–AH1562 and AH1610–AH1581 were used to amplify the effector cassette (Supplementary Table 2 and Supplementary Fig. 3). Another primer pair, AH1566–AH1615, that amplified the effector cassette and the flanking 5' *HS4* insulator tandem repeat sequences was used if the effector cassette could not be amplified with the more proximal primers. Owing to the highly repetitive 5' *HS4* sequence, the use of primers within the sequence or PCR through the sequence with external primers was not always reliable. PCR cycling conditions were: 30 s at 98 °C; 5 cycles of 10 s at 98 °C, 30 s at 58 °C, 2 min at 72 °C; 30 cycles

of 10 s at 98 °C, 30 s at 55 °C, 2 min at 72 °C; and a final extension for 2 min at 72 °C. If primer pairs flanking the driver or effector cassettes failed to generate amplicons, possibly due to large indel aberrations, TAIL-PCR was performed according to Liu and Whittier[50] using specific primers (SPs) SP1 (AH1557), SP2 (AH1635), SP3 (AH1638), and an arbitrary degenerate primer (AH1117). All PCR reactions and PCR product digestions resulting in the same profile or sequence were replicated a minimum of two or more times.

**Reporting summary.** Further information on research design is available in the Nature Research Reporting Summary linked to this article.

## Data availability

The data and materials that support the findings of this study are available from the corresponding author upon reasonable request. The *piggyBac* driver vector, pBXL {attP_PUbDsRed.T3_fa_s1-tTA_a}, annotated sequence is available from the NCBI GenBank database, accession no. MT453110.

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

## Acknowledgements

We thank Dr. Carsten Horn and Dr. Ernst Wimmer for sharing the M5.II lethal effector strain and the *s1-tTA* driver construct plasmid, and Dr. Rodney Nagoshi and Dr. Daniel Hahn for comments on the manuscript. This project was supported by the USDA-Biotechnology Risk Assessment Program competitive grant no. 2015-33522-24094 from the USDA National Institute of Food and Agriculture (to A.M.H.) and the Emmy Noether Program SCHE 1833-1/1 of the German Research Foundation (to M.F.S.). This study benefited from discussions at meetings for the Coordinated Research Project, "Comparing Rearing Efficiency and Competitiveness of Sterile Male Strains Produced by Genetic, Transgenic or Symbiont-based Technologies," funded by the International Atomic Energy Agency (IAEA).

## Author contributions

Y.Z. and A.M.H. designed experiments and analyzed data; Y.Z. and M.F.S. performed research; and all authors wrote the paper.

## Competing interests

The authors declare no competing interests.
