## [Peer Review File · Nature Communications]

Reviewers' Comments:

Reviewer #1:

Remarks to the Author:

I apologize for the delay in getting this back. I have read the authors rebuttal and their revised manuscript. There are a few places where i could quibble about the specific language used to discuss few points about gene drive and the evolution of resistance, but these are minor points where reasonable people may differ.

My one very minor point is that the authors should include a new reference, Knudsen, et al, published in G3, into their discussions regarding implications of their work on the evolution of resistance to tet-based lethality systems in insects.

REVIEWERS' COMMENTS:

Reviewer #1 (Remarks to the Author):

I apologize for the delay in getting this back. I have read the authors rebuttal and their revised manuscript. There are a few places where i could quibble about the specific language used to discuss few points about gene drive and the evolution of resistance, but these are minor points where reasonable people may differ.

My one very minor point is that the authors should include a new reference, Knudsen, et al, published in G3, into their discussions regarding implications of their work on the evolution of resistance to tet-based lethality systems in insects.

Bruce Hay

Response: Reference to the Knudsen et al. paper is now included in the context of discussions of inherent suppressors of lethality. Since the points in question about gene drive and the evolution of resistance are not specified, the reviewer's concerns cannot be addressed, though this is apparently a minor issue.

** See Nature Research's author and referees' website at www.nature.com/authors for information about policies, services and author benefits